# Embedding Synthetic Off-Policy Experience for Autonomous Driving via Zero-Shot Curricula

Eli Bronstein*     Sirish Srinivasan*     Supratik Paul*
Aman Sinha     Matthew O'Kelly     Payam Nikdel     Shimon Whiteson
Waymo, LLC
{ebronstein, sirishs, supratikpaul, thisisaman, mokelly, payamn, shimonw}@waymo.com

**Abstract:** ML-based motion planning is a promising approach to produce agents that exhibit complex behaviors, and automatically adapt to novel environments. In the context of autonomous driving, it is common to treat all available training data equally. However, this approach produces agents that do not perform robustly in safety-critical settings, an issue that cannot be addressed by simply adding more data to the training set—we show that an agent trained using only a 10% subset of the data performs just as well as an agent trained on the entire dataset. We present a method to predict the inherent difficulty of a driving situation given data collected from a fleet of autonomous vehicles deployed on public roads. We then demonstrate that this difficulty score can be used in a zero-shot transfer to generate curricula for an imitation-learning based planning agent. Compared to training on the entire unbiased training dataset, we show that prioritizing difficult driving scenarios both reduces collisions by 15% and increases route adherence by 14% in closed-loop evaluation, all while using only 10% of the training data.

**Keywords:** Imitation Learning, Curriculum Learning, Autonomous Driving

## 1    Introduction

Autonomous vehicles (AV) typically rely on optimization-based motion planning and control methods. These techniques involve bespoke components specific to the deployment region and AV hardware, and require copious hand-tuning to adapt to new environments. An alternative approach is to apply machine learning (ML) to the lifetimes of experience that AV fleets can collect within days or weeks. A paradigm shift to ML-based planning could automate the adaptation of behaviors to new areas, improve planning latency, and increase the impact of hardware acceleration.

For example, imitation learning (IL) can utilize the large tranches of expert demonstrations collected by the regular operations of AV fleets to produce policies that perform well in common scenarios, without the need to specify a reward function. However, both the distribution from which experiences are sampled and the policy used to generate the demonstrations can critically affect the IL policy's performance [1]. The training data distribution is especially important when learning methods are applied to problems characterized by long tail examples (*c.f.* [2, 3, 4, 5, 6]). In the case of autonomous driving, the vast majority of observed scenarios are simple enough to be navigated without any negative safety outcomes. A visual inspection of a random subset of our data suggests that half of it consists of scenarios with the AV as the only road user in motion, while another quarter contains other moving road users, but not necessarily close enough to the AV to affect its behavior. As a result, IL policies may not be robust in safety-critical and long tail situations.

Reinforcement learning (RL) can be used to explicitly penalize poor behavior, but due to the complex nature of driving [7], it is difficult to design a reward function for AVs that aligns with human expectations. Even if reward signals were provided for safety-critical events or traffic law violations (e.g., collisions, running a red light), they would be extremely sparse since such events are quite rare. Furthermore, exploration to collect more long tail data is challenging due to safety concerns [8].

Despite these issues, most learning-based robotics and AV applications use the naive strategy of creating training datasets from all available demonstrations. In the context of AVs, since most driving situations are simple, this strategy is both inefficient and unlikely to generate a policy that is robust

---

*Denotes equal contribution.

6th Conference on Robot Learning (CoRL 2022), Auckland, New Zealand.

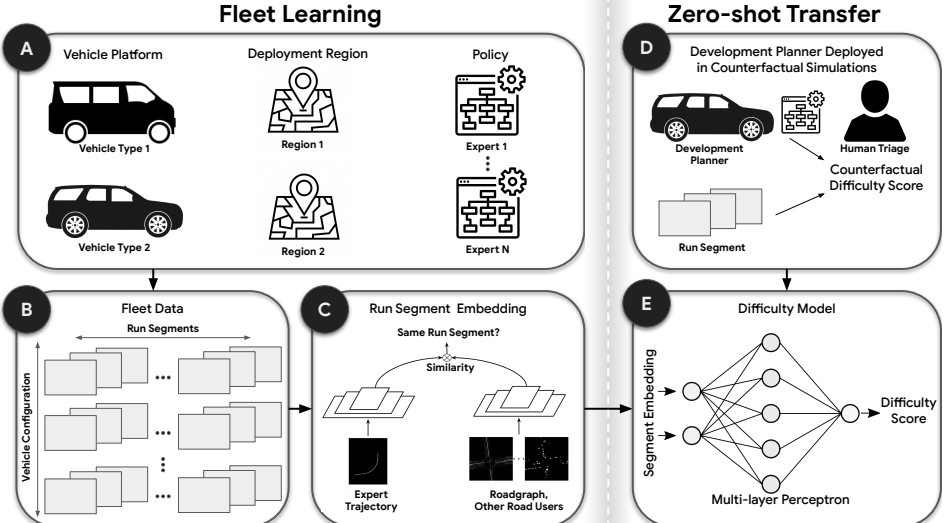

Figure 1: **A** The fleet collects experiences with a variety of policies in multiple operational design domains. **B** The fleet data is sharded into run segments. **C** Fleet data is used to learn an embedding that maps a run segment to a vector space based on similarity. **D** Run segments are selected for counterfactual simulations and human triage; the outcome of this process is a labeled set of difficulty scores. **E** An MLP is trained to regress from embeddings to the difficulty labels.

to difficult scenarios. A common solution is to upsample challenging examples, either by increasing their sampling probability by a predetermined factor [9] or with curriculum learning [10], i.e., dynamically updating the sampling probability during training based on the agent's performance.

However, both of these approaches include significant hurdles. Upsampling requires that we know which examples are part of the long tail *a priori*, as in standard classification problems where the class-label imbalance can inform a sampling strategy. In IL and RL, no such labels are available. As such, curriculum learning is more suitable since it uses the agent's current performance to identify hard examples. However, standard approaches to curriculum learning are specific to the agent being trained; they do not, for example, utilize data collected by deployed AVs running other planners, which can provide more general, policy-agnostic insights into the long tail of driving.

In this paper, we propose an approach (summarized in Figure 1) that addresses the challenges of upsampling and curriculum learning applied to an AV setting. Developing a road-ready AV generally involves both collecting real-world data with an *expert*, which can be a combination of human drivers and thoroughly evaluated AV planners; and evaluating new *development planners*, which are regularly simulated on the data collected by the expert to identify potential failure modes, generating a large counterfactual dataset. Our method uses this readily available data to train a *difficulty model* that scores the inherent difficulty of a given scenario by predicting the probability of collisions and near-misses in simulation. This difficulty model provides several key benefits: 1) it is computationally less expensive to predict a driving situation's difficulty than to simulate it for a policy being trained; 2) the model learns the inherent, policy-agnostic difficulty of a scenario because it is trained on multiple development planners in different geographic regions; and 3) the model predicts a continuous score that can be used to identify scenarios within an arbitrary difficulty range, rather than obtaining a few counterfactual failures.

We show that a zero-shot transfer of this model can identify long-tail examples that are difficult for a new IL-based planning agent—without any fine tuning. This allows us to upsample difficult training examples without expensive evaluation of the agent during training. Though we train the planning agent using IL as a case study, our approach can be applied to any ML-based planning approach. The main contributions of this paper are:

1. We train a model to predict which driving scenarios are difficult for development planners and show that it can zero-shot transfer to the task of finding challenging scenarios on which to train an ML-based planning agent. This generalization suggests that the model can predict the inherent difficulty of a driving situation.

2. We show that training an ML-based planning agent on unbiased driving data leads to poor performance on difficult examples since easy driving scenarios dominate rarer, harder cases.

3. We show that using our difficulty model to upsample more challenging scenarios reduces collisions by 15% and increases route adherence by 14% on the unbiased test set. This suggests that there are significant diminishing returns in adding common scenarios to the training dataset.

## 2   Related Work

The application of RL to the task of autonomous driving has received significant attention in recent years [11]; proposed methods span the gamut of methodologies and the AV stack itself. RL has been used to address a variety of problems including end-to-end motion planning, behavior generation, reward design, and even behavior prediction. In this work, we focus on imitation learning techniques [12], which avoid direct specification of a reward function, and rely instead on expert demonstrations. As a result, they can capture subtle human preferences and demonstrate impressive performance on a variety of robotics tasks. However, despite many attempts [13, 14, 15], IL and RL techniques still struggle with the long tail present in the driving task [4].

Like this work, Brys et al. [16] and Suay et al. [17] consider how to leverage potentially suboptimal demonstrations to improve the efficiency and robustness of learning. Unlike these works, we use offline methods to learn a model of each scenario's difficulty and bias the distribution that IL is performed on. This approach is similar to baselines [18, 19] inspired by Peters and Schaal [9]; however, unlike these works, our setting does not provide a reward signal for the proposed demonstrations. Instead, we use offline, off-policy simulations to learn a foundation model with which we can efficiently approximate a scenario's difficulty, which would have otherwise required expensive counterfactual simulations during training. Our approach sidesteps the inefficiencies of performing rollouts of the learnt policy on the entire dataset since inference using the difficulty model is computationally much cheaper than simulation. Similar techniques have also been proposed by Brown et al. [20]; however, they focus largely on situations with severely suboptimal demonstrations where the reward is specified. Similar problems have also been identified in offline RL [21]. Interestingly, Kumar et al. [22] identify the tight relationship between imitation learning and offline RL, noting the theoretical advantage of incorporating reward information in settings like autonomous driving which must avoid rare catastrophic failures. Our experiments provide empirical support for this insight.

Curriculum learning (CL) [10] is also closely related to this work. While not originally classified as such, methods like automatic domain randomization, prioritized experience replay [23], and AlphaGo's self-play [24, 25] have led to superhuman game-playing agents and breakthroughs in sim2real transfer [23, 26, 27]. CL methods solve for surrogate objectives rather than directly optimizing the final performance of the learner. They control which transitions are sampled, the behavior of other agents in an environment, the generation of initial states, or even the reward function. CL methods are also characterized by whether they are used on- or off-policy. For example, Uesato et al. [28] exploit low quality policies to obtain failures in an on-policy RL setting. Finding hard examples in the training data using this approach requires repeatedly generating rollouts for each expert trajectory in the dataset. Such an approach is computationally infeasible when operating at scale since the training datasets can have hundreds of thousands of real-world driving miles. Similar approaches known as hard-negative mining have been used in supervised learning settings [29]; like Uesato et al. [28] they evaluate the difficulty of examples online.

Instead, we consider variants of CL that exploit off-policy data. As in the on-policy case, the key problem is to determine which data is interesting. Off-policy compatible methods are also generally surrogate-based. For example, they can select for diversity [30], moderate difficulty [27], surprise [23], or learning progress [31]. Our approach is most similar to Akkaya et al. [27], but instead of performing expensive agent evaluation during training, we use off-policy data both to train a foundation model [32], which encodes experiences, and to classify the difficulty of an interaction. We also utilize large-scale real-world data and demonstrate that simpler curricula are effective.

## 3   Background

**Model-based Generative Adversarial Imitation Learning:** Behavior cloning (BC) [33, 13] is a naive imitation learning method that applies supervised learning to match the expert's conditional action distribution: $\arg\max_\theta \ \mathbb{E}_{s,a\sim\pi_E}[\log \ \pi_\theta(a|s)]$. BC policies may suffer from covariate shift,

resulting in quadratic worst-case error with respect to the time horizon [34]. To address this issue, generative adversarial imitation learning (GAIL) [35] formulates IL as an adversarial game between the policy $\pi_\theta$ and the discriminator $D_\omega$. The discriminator is trained to classify whether a given trajectory was sampled from $\pi_\theta$ (labeled 0) or from the expert demonstration (labeled 1), and the policy is trained to generate trajectories that are indistinguishable from demonstrations:

$$\arg \max_\theta \ \arg \min_\omega \mathbb{E}_{s,a\sim\pi_\theta}[\log \ D_\omega(s,a)] + \mathbb{E}_{s,a\sim\pi_E}[\log(1 - D_\omega(s,a))].$$

GAIL minimizes the gap in the joint distributions of states and actions $p(s,a)$ between the policy and the expert, resulting in linear error with respect to the time horizon [36]. However, GAIL relies on high variance policy gradient estimates because it uses an unknown dynamics model, making its objective function non-differentiable. In contrast, model-based GAIL (MGAIL) [37] uses differentiable dynamics in combination with the reparameterization trick [38] to reduce the variance of the policy gradient estimates.

## 4 Method

A key challenge in commercial AV development is to design an AV planner that can safely and efficiently navigate real-world settings while aligning with human expectations. At any given time, there may exist multiple *development planners* under evaluation. Iteratively improving an AV planner typically involves the following three steps. 1) *Data Collection:* Real-world data is collected by a fleet of vehicles in the operational area. 2) *Data-Driven Simulation:* A development planner is tested in simulation by having it control the data-collecting ego vehicle in a *run segment*, or a short snippet of recorded driving data. 3) *Evaluation and Improvement:* The development planner is evaluated on key metrics based on these simulations, with potential issues identified and addressed.

As mentioned in Section 1, we consider an ML-based approach to developing an AV planner from the ground up. One option is to use imitation learning to train a *planning agent*. Given an initial dataset of logged expert driving, a naive approach is to train the agent on the entire dataset. However, this means that challenging long tail segments are used only a few times during training, yielding a planning agent that has difficulty negotiating similar situations [2, 5, 6]. Thus, to improve our agent, we require a method to upsample these rare segments.

### 4.1 Difficulty Model

The key idea behind our method is to use the real-world run segments replayed in simulation with development planners to learn a *difficulty model* that predicts the difficulty of a logged segment, i.e., whether a development planner is likely to have a poor safety outcome in simulation. We train the difficulty model on simulations of multiple development planners in different geographic areas, so it can be seen as marginalizing over a diverse distribution of development planners. This makes the model more likely to be able to identify the inherent difficulty of a segment. In turn, this facilitates the zero-shot transfer from training on data from development planners to inferring difficulty for a substantially different planning agent. Intuitively, segments that development planners find difficult are likely to be difficult for the planning agent as well. Specifically, we use the difficulty model's scores to inform our upsampling strategy for training the planning agent.

The evaluation process for development planners typically involves large-scale simulations, with potentially problematic behaviors flagged for engineers to address. We train the difficulty model to predict collisions and near-misses attributable to the development planner, as opposed to other road users. This data is generated in the normal course of the AV planner development cycle, so no new training data is needed.

Since we want to marginalize out the idiosyncrasies of individual development planners, we model a simulation's safety outcome $y \in \{0, 1\}$ (1 if a collision or near-miss occurred, 0 otherwise) as a function of the logged run segment alone. The input to our model is a learned segment embedding from a separately trained model. Given a logged run segment, we collect static features (e.g., road/lane layouts, crosswalks, stop signs), dynamic features (e.g., positions and orientations of other road users over time), and kinematic information about the data-collecting ego vehicle. We use these features to generate two top-down images of the segment: one of the ego vehicle's trajectory, and another of the static features and other road users' trajectories. We encode each image into a dense $d$-dimensional embedding vector (as in [39]) using a CNN and contrastively train

a classifier (e.g., [40]) with cross-entropy loss to determine if two images are from the same run segment (see Figure 1c). Our difficulty model is an MLP that learns a function $f : \mathbb{R}^d \to [0, 1]$ mapping the embedding to the simulated safety outcome $y$. We trained this model using cross-entropy loss on a dataset of 5.6k positive and 80k negative examples. The number of negative examples was downsampled by multiple orders of magnitude since the prevalence of simulated collisions and near-misses is extremely low. The model produces uncalibrated scores by design, as trying to calibrate it to the extremely small unbiased prevalence rate of positive examples is numerically unstable.

## 4.2 Sampling Strategies

Given the long-tail nature of the difficulty scores (see Figure 2), it is natural to upsample difficult segments during training. A standard solution for upsampling in classification problems is to create separate datasets for each class, and then generate a batch by sampling a specified proportion from each dataset. Since this requires discretized classes, it cannot be applied to our real-valued difficulty scores. Moreover, due to the large training dataset size it is not scalable to upsample individual segments: the entire dataset cannot fit in memory and random access to individual examples from disk is incompatible with distributed file sharding of data. Instead, we partition the dataset into ten equally sized buckets, each corresponding to a decile of the data by difficulty scores, with up/downsampling achieved by assigning different sampling probabilities to each bucket. This enables us to efficiently generate batches on the fly (e.g., sampling a weighted batch of $k$ run segments requires minimal overhead over the $k$ constant-time accesses to the head pointers of each bucket). This decile-based bucketing also ensures that our method is agnostic to the model scores, which are uncalibrated.

We consider two training variants: 1) a fixed weighting scheme for each bucket, held constant throughout training, and 2) a schedule of weights for each bucket that changes as training progresses. Specifically, we use the following three sampling strategies. "Highest-10%" trains the agent only on the highest scoring bucket (i.e., on the segments with the highest 10% difficulty scores). "Uniform-10%" upsamples difficult segments by setting each bucket's sampling weight to the range of difficulty scores in that bucket (in the limit of infinite buckets, this approaches a uniform distribution over the difficulty scores). "Geometric-schedule-10%" implements a geometric progression of weights with each bucket weighted equally at the beginning of training and weighted proportional to its average difficulty score at the end of training (see Appendix 8.2 for further details). In Section 5.4 we compare the performance of these training variants against several baselines. Our variants are trained on only a 10% sample of available data.

## 5 Experiments

To prevent information leakage between the difficulty model and the planning agent, the former is trained on a dataset collected more than six months prior to the dataset for the latter. The training dataset for the planning agent consists of over 14k hours of driving logged by a fleet of vehicles. We split the data into 10 second run segments, resulting in over 5 million training segments. We also create two test sets, chronologically separate from the training set to prevent train-test leakage. The first *unbiased test set* is composed of 20k segments sampled uniformly from logged data. The second set consists of 10k segments with difficulty scores in the top one percentile of the training data's score distribution. The distributions of the difficulty model scores for the train set and unbiased test set both have long tails (see Figure 2) – scores above 0.85 account for only around 0.5% of the dataset.

As described in Section 4.2, we split the training dataset into 10 equal sized buckets based on the difficulty score deciles. 200 run segments are further split from each training bucket to obtain validation buckets for model selection. To highlight the effect of our training schemes on performance on segments of varying difficulty, we use the same bucketing approach for the unbiased test set as for the training set. We report the full, unbiased test set results by aggregating over all buckets.

## 5.1 Baselines

We report three baselines for comparison, which differ in their training data: "Baseline-all" is trained on the full dataset, "Baseline-10%" is trained on a uniformly randomly sampled 10% of the full dataset, and "Baseline-lowest-10%" is trained only on the bucket with the lowest difficulty scores.

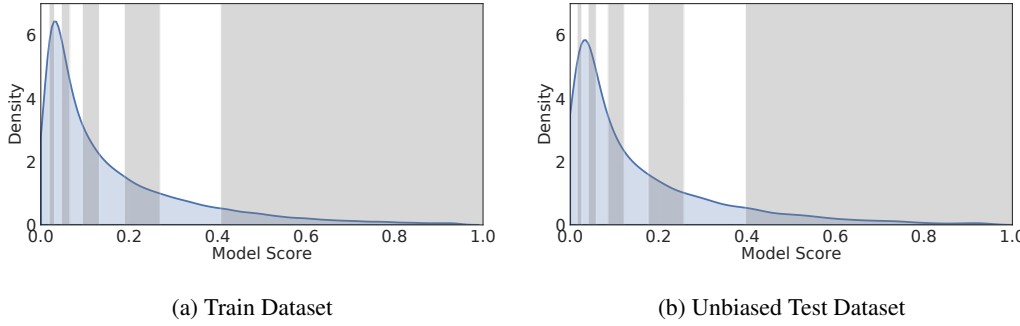

(a) Train Dataset        (b) Unbiased Test Dataset

Figure 2: Distribution of the difficulty model scores for the train and test datasets. The ten alternating shaded backgrounds indicate the thresholds of the decile buckets.

## 5.2 Training Details

We use the planning agent described in Bronstein et al. [41], which employs a stochastic continuous action policy conditioned on a goal route and is trained using a combination of MGAIL and BC. See Appendix 8.7 for additional details. We train 10 random seeds of each agent variant and baseline for 200k steps. After the initial 100k training steps, we evaluate each agent on the validation set at intervals of 10k steps. We select the agent checkpoint with the lowest sum of collision and off-road driving rates, and evaluate it on the held-out test set. Since the learnt policy is stochastic, we report the average performance of 16 independent rollouts for each test run segment.

## 5.3 Metrics

We assess the planning agent's performance using the following binary metrics (1 if the event of interest occurred in the segment, 0 otherwise):

1. *Route Failure*: the agent deviates from the goal "road route" at the start of the segment, which includes all lanes in the road containing the goal lane-specific route.

2. *Collision*: the agent's bounding box intersects with another road user's bounding box.

3. *Off-road*: the agent's bounding box exits the drivable road area.

4. *Route Progress ratio*: ratio of the distance traveled along the route by the agent and the expert.

We also report the overall failure rate as the union of the first three metrics; a segment is considered a failure if any of the binary metrics is nonzero. When comparing the performance of different agents, we prioritize this failure rate due to the safety-critical nature of driving, while also considering the route progress ratio to ensure the agents are making efficient forward progress.

## 5.4 Results

We present the performance of our training variants and baselines on the full, unbiased test set in Table 1. Each variant's action policy is conditioned on the expert's initial goal route, which is held constant throughout the segment.

We observe no significant difference between the performance of Baseline-10% and Baseline-all, demonstrating that simply increasing the training dataset size does not necessarily lead to better performance. Also, Baseline-lowest-10% has the worst performance for the collision and off-road metrics. This suggests that the easiest segments are not representative of the entire test set distribution and do not contain enough useful information to learn from. However, Baseline-lowest-10% achieves the lowest route failure rate. We believe this is because the least difficult training bucket is primarily composed of segments in which it is simple to follow the route, such as one-lane roads with no other road users and minimal interaction. This could cause the Baseline-lowest-10% agent to overfit to the route features and follow the route well at the expense of safety.

All three of our upsampling variants achieve significantly lower collision rates, and comparable off-road and route failure rates to the baselines (with the exception of Baseline-lowest-10%'s route failure rate). This key result demonstrates that segments with high predicted difficulty contain the

Table 1: Evaluation of agents and baselines on the full unbiased test set (mean ± standard error of each metric across 10 seeds). For all metrics except route progress, lower is better.

| Agent Variant | Route Failure rate (%) | Collision rate (%) | Off-road rate (%) | Route Progress ratio (%) | Failure rate (%) |
|---|---|---|---|---|---|
| Baseline-all | $1.38_{\pm 0.13}$ | $1.46_{\pm 0.09}$ | $\mathbf{0.73}_{\pm \mathbf{0.07}}$ | $81.21_{\pm 0.39}$ | $3.33_{\pm 0.20}$ |
| Baseline-10% | $1.34_{\pm 0.06}$ | $1.50_{\pm 0.09}$ | $\mathbf{0.67}_{\pm \mathbf{0.06}}$ | $\mathbf{81.12}_{\pm \mathbf{0.37}}$ | $3.28_{\pm 0.13}$ |
| Baseline-lowest-10% | $\mathbf{1.14}_{\pm \mathbf{0.05}}$ | $4.15_{\pm 0.11}$ | $0.98_{\pm 0.10}$ | $\mathbf{81.88}_{\pm \mathbf{0.41}}$ | $5.91_{\pm 0.13}$ |
| Highest-10% | $1.33_{\pm 0.06}$ | $\mathbf{1.23}_{\pm \mathbf{0.09}}$ | $0.74_{\pm 0.02}$ | $77.95_{\pm 1.33}$ | $\mathbf{3.10}_{\pm \mathbf{0.10}}$ |
| Uniform-10% | $1.35_{\pm 0.09}$ | $\mathbf{1.17}_{\pm \mathbf{0.08}}$ | $0.75_{\pm 0.07}$ | $80.67_{\pm 0.73}$ | $\mathbf{3.07}_{\pm \mathbf{0.17}}$ |
| Geometric-schedule-10% | $\mathbf{1.19}_{\pm \mathbf{0.07}}$ | $\mathbf{1.25}_{\pm \mathbf{0.04}}$ | $\mathbf{0.74}_{\pm \mathbf{0.10}}$ | $80.48_{\pm 0.36}$ | $\mathbf{2.92}_{\pm \mathbf{0.11}}$ |

majority of useful information needed for good aggregate performance. Geometric-schedule-10% has the largest improvement over the baselines, with a significantly lower collision rate, comparable route failure and off-road rates, and a minimal decrease in the route progress ratio. This highlights the advantage of observing the whole spectrum of data at the start of training, and progressively increasing the proportion of difficult segments to emphasize more useful demonstrations.

To get a more nuanced view of each variant's performance, we compare the variants to Baseline-10% for each of the test buckets. Figure 3 shows the performance for the lowest (0-10%), low/mid (30-40%), highest (90-100%), and long tail (99-100%) test buckets. See Figures 5 and 6 in the Appendix for metrics for all the test buckets.

Not only does each agent's collision rate correlate with the difficulty score, but so do the route failure and off-road rates, with the exception of Highest-10%'s off-road rate. This shows that segments that were challenging for development planners are also likely to be challenging for our planning agent, which enables the zero-shot transfer of the difficulty model. It also demonstrates that although the difficulty model was only trained to predict collisions and near-misses, its predicted score describes a broader notion of difficulty, as measured by other key planning metrics.

On the highest and long tail buckets, Highest-10% and Uniform-10% achieve much lower collision rates and overall failure rates than Geometric-schedule-10% and the baseline. This shows that upsampling difficult segments results in better *overall* performance on those segments, not just on metrics that are highly correlated with the difficulty label (i.e., collisions and near-misses). This is encouraging, since it suggests that the training labels for the difficulty model do not need to fully define expert driving behavior in order for the resulting planning agent to exhibit improved performance across multiple metrics. However, Highest-10% and Uniform-10% perform comparable to, or worse than the baseline on the lowest and low/mid buckets across all metrics, with especially poor performance on the route failure and off-road metrics. Thus, extreme upsampling of difficult segments sacrifices performance at the other end of the spectrum, since the easiest segments become too rare in the training data. Geometric-schedule-10% addresses this issue by upsampling difficult segments while maintaining sufficiently broad coverage over the difficulty distribution. While it does not achieve equally low collision rates as the Highest-10% and Uniform-10% variants on the highest and long tail buckets, it outperforms the baseline on collisions and performs well on the lowest and low/mid buckets, yielding the best overall performance.

## 6 Limitations

While our difficulty model successfully identified challenging segments, it was only trained to predict collisions and near-misses, which are just one indication of difficulty. There are other labels that would be helpful for a more comprehensive difficulty model, such as traffic law violations, route progress, and discomfort caused to both the ego vehicle's passengers and other road users. Moreover, the difficulty model could be improved by incorporating the severity of the negative safety outcome into the training labels. Furthermore, as noted in Section 1, a large proportion of the available data consists of situations with very few other road users in the scene. The difficulty model could be replaced with a heuristics-based approach of pruning such scenarios, though the viability of doing so is difficult to gauge *a priori*.

In terms of evaluation metrics, we focused primarily on safety metrics, since these are of paramount importance for real-world deployment. However, we have not considered other facets of driving like comfort and reliability, which can also significantly affect the viability of ML-based planners.

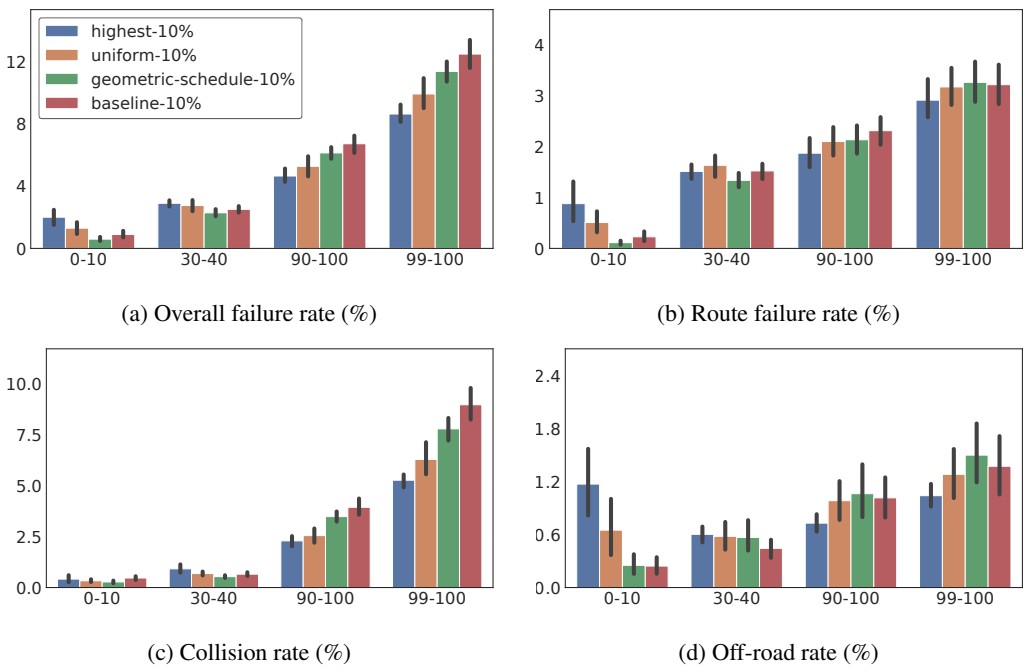

(a) Overall failure rate (%)

(b) Route failure rate (%)

(c) Collision rate (%)

(d) Off-road rate (%)

Figure 3: Metrics for the Baseline-10%, Highest-10%, Geometric-schedule-10%, and Baseline-10% variants on multiple decile test buckets according to the difficulty score. For each metric, each variant's performance is shown for the lowest (0-10%), low/mid (30-40%), highest (90-100%), and long tail (99-100%) test buckets.

Finally, while we have demonstrated that our method of upsampling long tail segments leads to better performance, we have done so only for an agent trained using MGAIL. Quantifying the performance gains with other learning methods remains a topic for future work.

# 7 Conclusion

We showed that the naive strategy of training on an unbiased driving dataset is suboptimal due to the large fraction of data that does not provide additional useful experience. By utilizing readily available data collected while evaluating development planners in simulation, we trained a model to identify difficult segments with poor safety outcomes. We then applied this model in a zero-shot manner to develop training curricula that upsample difficult examples. Planning agents trained with these curricula outperform the naive strategy in aggregate and are more robust in challenging, long tail scenarios. However, overly aggressive upsampling produces policies that do not handle simpler situations well. We conclude that sampling strategies that prioritize difficult segments but also include easier ones are likely to achieve the best overall performance.

We have also showed that training on the full dataset does not yield any significant benefit over training on only 10% of the data sampled uniformly at random, demonstrating that simply adding more unbiased data to the training set does not necessarily improve performance. This suggests that we can use our difficulty model to reduce the cost of AV system development in two areas: targeting active data collection when operating a fleet of vehicles and selective retention of large-scale sensor logs. Namely, since the difficulty model can predict which driving scenarios are likely to be challenging for new planning agents, we could identify geographic "hotspots" where these scenarios occur and use these locations to inform our data collection process. Furthermore, since biasing the planning agent's training dataset toward difficult segments leads to better results with only a fraction of the available data, we could use the difficulty model scores to reduce the amount of stored data without sacrificing downstream performance.

**Acknowledgments**

We thank Ben Sapp, Eugene Ie, Jonathan Bingham, and Ryan Polkowski for their helpful comments, and to Ury Zhilinsky for his support with experiments and infrastructure.

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
