# OpenReview forum: "Embedding Synthetic Off-Policy Experience for Autonomous Driving via Zero-Shot Curricula"
_robot-learning.org/CoRL/2022/Conference — CoRL 2022 Oral_

### Official Review · Reviewer_Dcm5 · 2022-07-18

**Originality:** Fair
**Technical Quality:** Good
**Clarity Of Presentation:** Fair
**Impact:** 3

**Recommendation:**

Weak Accept: I recommend accepting the paper, but will not argue for my recommendation if the majority of other reviewers have a different opinion.

**Summary:**

This work presents a method to assess the difficulty of a scenario in an autonomous driving data set. Instead of using the full data set for motion planning in an imitation learning setting, the proposed method only considers a subset of the data set which is assessed as difficult to generate curricula. These curricula are used to train an imitation learning agent for motion planning. Experiments show that the proposed method achieves improved performance in various criteria such as reduced collisions in the autonomous driving setup.

**Issues:**

- While reading the text I find it quite hard to follow which parts of the framework are optimized. I believe the reader would appreciate it if the objectives which are optimized are explicitly mentioned in the text in the sense that also the optimized models/distributions have an explicit mathematical description. By these definitions the text can also easily refer to the parts and explaining will probably be easier

- While the evaluations of the paper are thorough and nice, I was wondering if a comparison to other similar works (e.g. [16, 17]) (as stated in the related work part) would make sense. Is there a specific reason why they were not conducted?

- Does the sentence in lines 94-96 make sense? It is mentioned that a "counterfactual simulation" is used, but later in the same sentence it is mentioned that "... would have otherwise required expensive counterfactual simulation". Probably I missed something and got confused with this sentence.

**Quality Of The Limitations Section:**

Limitations are addressed clearly

**Reviewer Expertise:**

2: The reviewer is willing to defend the evaluation, but it is quite likely that the reviewer did not understand central parts of the paper

**Robotics Focus:**

Highly relevant to robotics but no hardware experiments

**Strengths And Weaknesses:**

The paper provides thorough evaluations of the sampling techniques and provides a nice related work part.
However, I find it quite hard to follow which parts are optimized with only the text (see issues)
Additionally, a comparison to other baselines which use different techniques e.g. [16, 17] would make sense for me.



**Summary Of Recommendation:**

Based on my concerns and issues I recommend assessing a weak reject. However, as I am not an expert in this field, I am fine if the other reviewers think that the current version of the paper is worth being published.

# Post Rebuttal decision:
Based on the answers of the authors, I would like to raise my recommendation from weak reject to **weak accept**.

---

### Official Review · Reviewer_fmkD · 2022-07-24

**Originality:** Excellent
**Technical Quality:** Excellent
**Clarity Of Presentation:** Excellent
**Impact:** 4

**Recommendation:**

Strong Accept: I recommend accepting the paper and will argue for my recommendation even if other reviewers hold a different opinion.

**Summary:**

This work presents a method for predicting situational difficulty, given data collected from autonomous vehicles deployed on public roads, then uses the predictor to generate curricula for a planning-based imitation learner.

**Issues:**

* It would be helpful to understand the essential components of the foundation difficulty model, through ablation experiments.

**Quality Of The Limitations Section:**

Limitations are addressed clearly

**Reviewer Expertise:**

5: The reviewer is absolutely certain that the evaluation is correct and very familiar with the relevant literature

**Robotics Focus:**

Highly relevant to robotics but no hardware experiments

**Strengths And Weaknesses:**

## Strengths
* Strong motivation. The particular choice of underlying data distribution driving imitation learning critically affects performance, particularly in safety critical scenarios with long tail events. This work adds to the body of work that shows just how large an affect changing the data distribution can have on outcomes. In particular this highlights how the common recent approach of "just scaling up data" can have diminishing returns. Although this is demonstrated in the autonomous driving domain, I think these findings should be quite relevant to any other where easy examples dominate much rarer safety critical events. Furthermore, L53-L65 do a good job of motivating the particular choice of a machine-learned "foundation difficulty model" over the more expensive conventional approach of using planners to generate counterfactual failure datasets.
* Clear method description. Good description of the full inputs and outputs to the foundation difficulty model and how it was trained, as well as how each part of the overall system fit together.
* Careful experiments highlight a surprising and important point: simply increasing the training dataset size does not necessarily lead to better performance. Prioritizing difficult scenarios during imitation learning, using the presented method, allows a model to be trained on 10% of the full dataset, that behaves competitively with one trained on the full dataset. This highlights a weakness in the popular "just scale up data" approach. I think the magnitude of the improvement in sample efficiency via this prioritization scheme could be an important finding for others working on autonomous driving.
* Good discussion of related work, highlighting how this off-policy curricula approach compares to on-policy versions.
Interesting study (4.2) comparing simpler fixed weighting amongst difficulty buckets to more complex scheduled weighting.
* Careful seeded experiment design which help assess statistical significance.

## Weaknesses
* No ablations on foundation model. While the benefits of employing a "foundation difficulty model" seem clear from the experiment, the reader is not left with too much insight on which elements of this foundation model are the most essential. For example, L159 the model is trained on "multiple development planners in different geographic areas". This is claimed as the key to generalization, but we do not know *how important* it is. It would be interesting to see for example, how much data is required to train a sufficient foundation difficulty model, whether multiple regions of data or data coming from multiple planners are necessary, etc.


**Summary Of Recommendation:**

This work provides experiments highlighting the surprising diminishing returns "just scaling up data" can have in learning autonomous driving. It presents a new "foundation difficulty model" approach, trained on real offline driving data, which is used to drive a reweighting curriculum imitation learning.  Careful experiments show that this system learns controllers that are competitive with those receiving 10x more data. I think this is a concrete result that others doing large scale imitation learning will benefit from.

---

### Official Review · Reviewer_4k9b · 2022-07-31

**Originality:** Fair
**Technical Quality:** Fair
**Clarity Of Presentation:** Good
**Impact:** 2

**Recommendation:**

Strong Reject: I recommend rejecting the paper and will argue for my recommendation even if other reviewers hold a different opinion.

**Summary:**

This paper proposes a system to label the difficulty of collected driving cases in a policy-agnostic manner so that this system can be used to pick safety-critical training data for ML-based planning modules. By changing the training data distribution according to the labeled difficult score, the performance of the IL policy can be improved. It has a lower failure rate in safety-critical scenarios, compared to the agent trained on unbiased training data.

**Issues:**

Which dataset is used in this work? Is it an open dataset or an in-house one?

**Quality Of The Limitations Section:**

Limitations are addressed clearly

**Reviewer Expertise:**

4: The reviewer is confident but not absolutely certain that the evaluation is correct

**Robotics Focus:**

Relevant but unlikely to deploy to hardware in near future

**Strengths And Weaknesses:**

**Strength**:
The motivation for this work is important. Ways to tame the long-tail distribution should be developed for acquiring a learning-based controller. This work proposes a new approach to achieve this by giving a difficulty score.

**Weakness**:
1. It is well known that changing the training data distribution can improve performance.
2. The description of the method is ambiguous and difficult to understand. Consider adding a figure to better illustrate it.
3. In the experiment, other policy-agnostic baselines, which can also score the difficulty of driving scenarios should be included as baselines, such as RSS (Mobile Eye *x* Intel).

Personally, I do not believe that training ML-policy in replayed cases is a good approach for autonomous driving, especially for solving safety-critical cases where rich interaction is required. Since the replayed vehicles can not react to the ego vehicles, the policy can not be adapted to the real world. We should consider generating some realistic interactive cases from the collected data for learning-based policy training and bridging the sim-2-real gap.

**Summary Of Recommendation:**

The method description is not accurate. Notations and figures can be used to improve the quality. Also, more evidence should be provided to point out why this scoring method is better than other rule-based ones, such as RSS.

---

### Official Review · Reviewer_CdL1 · 2022-07-31

**Originality:** Good
**Technical Quality:** Very Good
**Clarity Of Presentation:** Excellent
**Impact:** 4

**Recommendation:**

Weak Accept: I recommend accepting the paper, but will not argue for my recommendation if the majority of other reviewers have a different opinion.

**Summary:**

The authors argue as follows:

Naively training on all driving data via imitation learning is suboptimal since some examples are harder than others. A way to solve this is via upsampling/upweighting more difficult examples, and the main question is how one would determine this difficulty metric in an efficient way. We can design a separate model which predicts difficulty as a function of run segments, where the label is provided via simulation and simulated / ground-truth metrics of risk. This difficulty model is then run over the entire dataset and predicts a difficulty score for each run segment. This then allows for training of imitation learning with upsampling of the more difficult scenarios and with a curriculum of training on easier->harder examples, which is shown to improve results compared to equal weighting of data.


**Issues:**

See questions in weaknesses


**Quality Of The Limitations Section:**

Limitations are addressed clearly

**Reviewer Expertise:**

4: The reviewer is confident but not absolutely certain that the evaluation is correct

**Robotics Focus:**

Highly relevant to robotics but no hardware experiments

**Strengths And Weaknesses:**

Strengths
+ The authors present a clear way to predict driving difficulty via counterfactual simulation and explain the method clearly
+ The data sampling methods do demonstrate zero-shot transfer, which implies that using difficulty based on counterfactual simulation does improve imitation learning of AV planning, based on the provided metrics.
+ Several strategies for sampling are evaluated, which include a curriculum schedule, and performance is visualized which demonstrate tradeoffs between performance on easy and hard data
+ The ablations of the upsampling strategy and curriculum strategy against the baseline which indicate that the difficulty score is a useful method of upsampling against uniform weighting of the data
+ The authors provide more detailed experiments in the appendix, including adaptive importance sampling, which help understand the limits of these upsampling methods
+ The visualizations in the provided video are helpful

Weaknesses
- The baseline is the naive method, and there are no comparisons to any prior work. Albeit, relevant prior works may be hard to compare against; the authors mention closest work are on-policy
- There could be some discussion of or attempts at the feasibility of using real data instead of simulated counterfactual examples for training the difficulty model. Or, as an ablation, could some difficulty heuristic be computed for each imitation learning run-segment instead of using the difficulty model?
- One of the key aspects of this work is the efficiency advantage compared to on-policy difficulty computation methods. The tradeoffs/feasibility here could be better understood if all the dataset sizes were provided.
- There could be some discussion about using some weighting term based on the difficulty score and uniform sampling, rather than sampling the dataset into buckets. The authors mention that the scores are uncalibrated, but some kind of bucketing of a weighting term may resolve this similarly to what is used in the paper. This may allow for more hyperparameter search compared to the current method of fixed buckets which presents infrastructure challenges.
- How should we understand the tradeoffs between the metrics, e.g. is failure rate or route progress ratio more important? It seems that the two are somewhat inversely correlated.


**Summary Of Recommendation:**

The work provides compelling motivations and results for using difficulty score based on synthetic off-policy data in imitation learning planning. It is clear how such a method could improve AV planning, and how it could improve training such methods at large dataset scale, where other kinds of difficulty metrics may not be as feasible. Therefore, it seems that this work has the potential for larger impact. There are still some limitations in the set of works or alternative design choices compared against and understanding of the datasets and metrics used.

---

### Meta-Review · Area_Chair_HdCk · 2022-08-31

**Recommendation:** Accept (Oral)
**Confidence:** 4

**Metareview:**

This paper describes an approach for upsampling a large dataset of driving scenarios in order to enable a learning agent to face the most difficult, safety critical scenarios during learning. It enables the learning agent to reduce collisions and increase route adherence, using a small fraction of the training data.

Strengths:

* The motivation for this work is important as it is a way to tame the long-tail distribution.
* This work uses real data from real autonomous driving logs making it highly relevant to real-world problems.
* Extensive experiments that highlight an unintuitive point: increasing the training data set size does not necessarily increase performance.

Weaknesses:

* Replaying cases from logs is challenging as the vehicles in the log cannot react to the new policy without further simulation.